# Prevalence of bovine tuberculosis and characterization of the members of the *Mycobacterium tuberculosis* complex from slaughtered cattle in Rwanda

**Jean Bosco Ntivuguruzwa**[1,2☯]*, Anita L. Michel[1☯], Francis Babaman Kolo[1‡], Ivan Emil Mwikarago[3¤‡], Jean Claude Semuto Ngabonziza[4,5‡], Henriette van Heerden[1☯]

**1** Bovine Tuberculosis and Brucellosis Research Programme, Department of Veterinary Tropical Diseases, Faculty of Veterinary Science, University of Pretoria; Pretoria, South Africa, **2** Department of Veterinary Medicine, College of Veterinary Medicine, University of Rwanda, Kigali, Rwanda, **3** Department of Human Medicine and Device assessment and Registration, Rwanda Food and Drug Administration, Kigali, Rwanda, **4** National Reference Laboratory Division, Department of Biomedical Services, Rwanda Biomedical Centre, Kigali, Rwanda, **5** Department of Clinical Biology, School of Medicine and Pharmacy, College of Medicine and Health Sciences, University of Rwanda, Kigali, Rwanda

☯ These authors contributed equally to this work.
¤ Current address: Department of Human Medicine and Device assessment and Registration, Division of Human Medicine, Assessment and registration, Rwanda Food and Drug Administration, Kigali, Rwanda
‡ AB and CD also contributed equally to this work.
* boscus2@gmail.com

## Abstract

### Background

Bovine tuberculosis (bTB) is an endemic disease in Rwanda, but little is known about its prevalence and causative mycobacterial species. The disease causes tremendous losses in livestock and wildlife and remains a significant threat to public health.

### Materials and methods

A cross-sectional study employing a systematic random sampling of cattle (n = 300) with the collection of retropharyngeal lymph nodes and tonsils (n = 300) irrespective of granulomatous lesions was carried out in six abattoirs to investigate the prevalence and identify mycobacterial species using culture, acid-fast bacteria staining, polymerase chain reaction, and GeneXpert assay. Individual risk factors and the origin of samples were analysed for association with the prevalence.

### Findings

Of the 300 sample pools, six were collected with visible TB-like lesions. Our findings demonstrated the presence of *Mycobacterium tuberculosis* complex (MTBC) in 1.7% (5/300) of sampled slaughtered cattle. *Mycobacterium bovis* was isolated from 1.3% (4/300) animals while one case was caused by a rifampicin-resistant (RR) *M. tuberculosis*. Non-tuberculous mycobacteria were identified in 12.0% (36/300) of the sampled cattle. There were no

**Data Availability Statement:** All relevant data are within the manuscript and its Supporting Information files.

**Funding:** This research was funded by grant FA DGD-ITM 2017 – 2021 awarded to HvH by the Belgian Directorate-General for Development Cooperation, through its Framework Agreement with the Institute of Tropical Medicine (https://www.itg.be). The funders had no role in study design, data collection and analysis, decision to publish, or preparation of the manuscript.

**Competing interests:** The authors have declared that no competing interests exist.

significant associations between the prevalence and abattoir category, age, sex, and breeds of slaughtered cattle.

## Conclusions

This study is the first in Rwanda to isolate both *M. bovis* and RR *M. tuberculosis* in slaughtered cattle indicating that bTB is present in Rwanda with a low prevalence. The isolation of RR *M. tuberculosis* from cattle indicates possible zooanthroponotic transmission of *M. tuberculosis* and close human-cattle contact. To protect humans against occupational zoonotic diseases, it is essential to control bTB in cattle and raise the awareness among all occupational groups as well as reinforce biosafety at the farm level and in the abattoirs.

### Author summary

Tuberculosis in cattle (bTB) causes financial losses to livestock owners and is a disease transmissible to humans especially those with an occupational risk through exposure to infected animals and animal products. This study aimed to identify the prevalence of bTB and characterize the mycobacterial species from cattle slaughtered in the six abattoirs in Rwanda. Four *M. bovis*, as well as one rifampicin-resistant (RR) *M. tuberculosis*, were identified from slaughtered cattle and, thus, the apparent bTB prevalence was 1.7% (5/300). Likely, the RR *M. tuberculosis* isolate was mostly likely of human origin and transmitted to cattle during close human-cattle contact. It is therefore essential to control bTB in cattle and reinforce the protection of farmworkers and abattoir workers who are always exposed to infected animals.

## Introduction

Apart from *Mycobacterium leprae*, the genus *Mycobacterium* comprises two groups, *Mycobacterium tuberculosis* complex (MTBC) and non-tuberculous mycobacteria (NTM) also known as atypical mycobacteria or mycobacteria other than tuberculosis (MOTT) [1]. The bovine tuberculosis (bTB) is a mycobacterial disease of cattle, other domestic and wild animals, as well as humans [2–5]. The disease is characterized by granulomatous lesions in affected tissues [2,4,6]. The disease is primarily caused by *Mycobacterium bovis* [7], however, other members of the MTBC have also been identified in diseased cattle such as *M. africanum* [8], *M. caprae* [9–11], *M. tuberculosis* [12,13] and *M. orygis* [14]. *Mycobacterium tuberculosis* infection in cattle has been identified more recently from African countries and is of concern [4,15–17]. These species belong to the MTBC whose members share 99.9% of their genome [18].

In Rwanda, only two studies are available on bTB prevalence including one that reported 0.5% prevalence in slaughtered cattle at société des abattoirs de Nyabugogo (SABAN Nyabugogo) [19]. This study also found that the disease was associated with financial losses related to the condemnation of carcasses which negatively affected the livelihood of small Rwandan farmers [19]. The other was a retrospective study recording TB-like macroscopic lesions at the same abattoir from 2006 to 2010 and reported the prevalence ranging from 1.4% in Kigali city to 11.8% in Eastern Province [20]. In Rwanda, the veterinary services lack the capacity and facility to isolate airborne pathogens. Thus, the control program for bTB relies mostly on monthly reports of gross TB-like lesions from the main private abattoir, SABAN Nyabugogo.

The cattle population in Rwanda was estimated at 1,293,768 in 2018 [21]. Although informal slaughtering of goats, sheep, chicken, and rabbits for family or small bar consumption does occur in Rwanda, it is estimated that 95% of slaughtered cattle are processed by abattoirs. Determining the bTB prevalence and identification of MTBC members is essential to understand the transmission dynamics at the animal-human interface and to design adequate control programs. The objective of this study was therefore to determine the prevalence of bTB and characterize MTBC members in slaughtered cattle in Rwanda as well as to investigate association with individual risk factors and the origin of samples. The findings of this study will contribute to building the bTB database essential for policymakers to establish informed control policies and strategies to mitigate bTB in Rwanda.

## Materials and methods

### Ethics statement

The authorization to conduct the study was obtained from the research screening and ethical clearance committee of the College of Agriculture, Animal Sciences and Veterinary Medicine, University of Rwanda (Ref: 026/DRIPGS/2017), institutional review board of the College of Medicine and Health Sciences, University of Rwanda (N° 006/CMHS IRB/2020), and Animal Ethics Committee of the Faculty of Veterinary Science, University of Pretoria, South Africa (V004/2020). Informed consents were obtained from district officials, managers of abattoirs, and owners of animals at the abattoirs.

### Study area

The present study was carried out in six abattoirs in Rwanda. Rwanda is a member of the East African Community (EAC) located in the southern hemisphere, near the equator. Six abattoirs that consented to participate in this study included high throughput abattoirs (n = 4) slaughtering more than 50 cattle daily, and low throughput abattoirs (n = 2), slaughtering 50 or less every day. Three of these abattoirs slaughtered cattle and goats and three specialised in cattle. The location of the six abattoirs is shown in Fig 1.

### Study design and sample size

A cross-sectional study was carried out from August 2018 through October 2019 to determine the prevalence of bTB and characterize *Mycobacterium* spp. in cattle slaughtered at abattoirs. The abattoirs that accepted to participate in the study were purposively selected based on their strategic locations in the thirty districts of Rwanda (Fig 1) and their slaughtering capacity. High throughput abattoirs received cattle from different districts. For instance, cattle that were sampled at SABAN abattoir located in Kigali City were from 19 districts including Nyarugenge, and Gasabo (Kigali City), Ngoma, Kirehe, Nyagatare, Gatsibo, Bugesera, and Kayonza (Eastern Province), Gakenke, Burera, Rulindo, and Gicumbi (Northern Province), Rutsiro, Karongi, Ngororero (Western Province), Ruhango, Nyanza, Kamonyi, and Muhanga (Southern Province). Cattle that were sampled at Rugano abattoir located in Kigali City were from four districts including Gasabo, Kicukiro, Nyarugenge (Kigali City), and Rwamagana of Eastern Province. Cattle that were sampled at Kamembe abattoir located in the Western Province, were from eight districts including Gisagara, Huye, Nyaruguru, Ruhango, Nyanza, Nyamagabe (Southern Province), Nyamasheke, and Rusizi (Western Province). Cattle that were sampled at Rubavu abattoir located in the Western Province, were from two districts including Nyabihu, and Rubavu (Western Province).

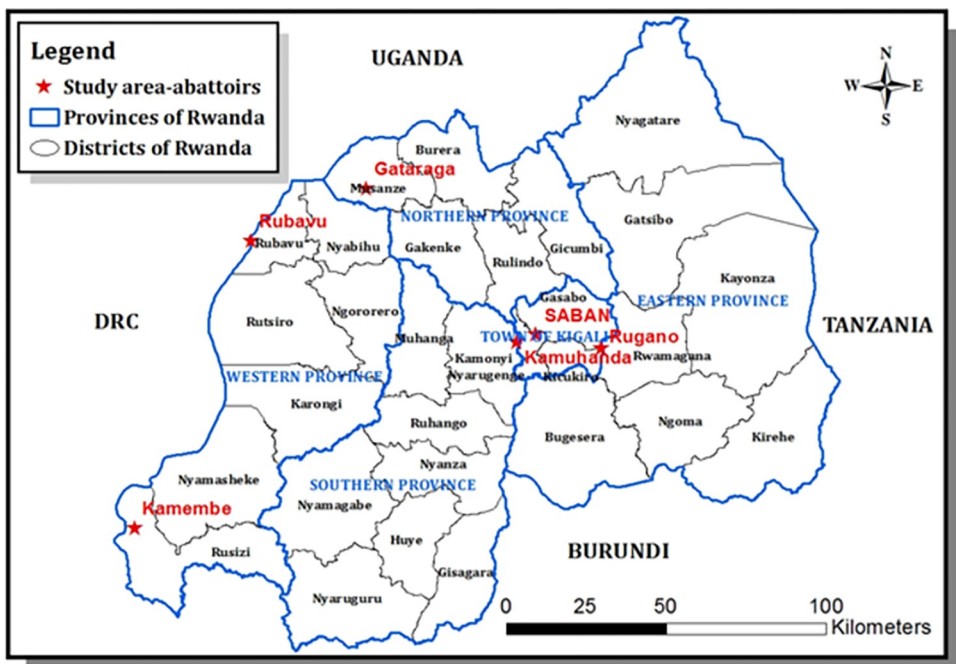

**Fig 1. Map of Rwanda with provinces and districts and red stars show the locations of abattoirs visited in this study.** This map was produced in the ArcGis Desketop 10.1–10.8.2 of the University of Rwanda with concurrent use and Education licence. Spatial data (shapefiles) are freely available from DIVA-GIS. (https://gadm.org/download_country.html and https://diva-gis.org/gdata).

The sample size was calculated as previously described [22].

$$N = \frac{Z^2 P(1 - P)}{d^2}$$

Where N is the sample size, a 95% confidence level (z) equivalent to 1.96; P is an expected prevalence (= 0.5%) based on a previous study in cattle in Rwanda [19], and the absolute precision (d = P/2) of 0.25%. According to the formula, the total sample size was supposed to be 291 but it was rounded to 300 cattle to include and respect the systematic sampling of 10 cattle for each of the 30 districts of Rwanda.

## Sampling procedure and tissue collection

A systematic random sampling procedure among dairy cattle was adopted, and the strategy was to sample five animals from the same district per day. The movement permit was collected on arrival to record the origin of the animals. The age was determined using dentition as previously described [23]. Animals of two years and above were selected with a preference on dairy except for some low throughput abattoirs that slaughtered mostly young male cattle. Animals were aligned in a crush pen and every fourth animal was selected, individual demographic information (district of origin, age, breed, and sex) recorded, restrained, marked on the head with original paint, and released for resting waiting for slaughter. Upon decapitation, the marked head was inspected, and medial and lateral retropharyngeal lymph nodes and lingual and palatine tonsils were aseptically collected into the same sterile 50 ml tube (pooled) and stored at—20˚C until processing.

## Culture and detection of acid-fast bacilli (AFB)

Retropharyngeal lymph nodes and tonsils from each sampled animal (n = 300) were decontaminated as previously described [24]. Briefly, tissues were thawed, sliced, grounded in a sterile mortar with sea sand (Glentham Life Sciences, UK). An aliquot of each pool of tissue samples was kept at -20˚C. Another aliquot was divided into two 50 ml falcon tubes. The content of one tube was decontaminated by adding an equal volume of 2% hydrochloric acid and the other one with an equal volume of 4% sodium hydroxide for 10 min at room temperature, then centrifuged at 3500 rpm for 10 min. The pellet was neutralized with 25 ml of sterile water centrifuged at 3500 rpm for 10 min [24]. Löwenstein-Jensen (LJ) medium was prepared according to the manufacturer's guidelines (Becton Dinkson, Franklin lakes, USA). Briefly, 37.3 g of LJ medium were completely dissolved in 600 ml of distilled water containing 12 ml of glycerol by heating. The dissolved LJ solution was then autoclaved at 121˚C for 15 min. The homogenised sterile egg content (1000 ml) was added to the cooled LJ solution, then, 7 ml of the mixture was distributed into 14 ml–tubes (slopes) which were immediately inspissated at 85˚C for 45 min. The slopes were checked for sterility and stored in the fridge until use. Glycerol was replaced by 4 g of sodium pyruvate (Labkem, Spain) for the preparation of LJ with pyruvate. The pellet that was decontaminated by hydrochloric acid and sodium hydroxide were each inoculated onto duplicate slopes of LJ with glycerol and duplicate slopes of LJ with sodium pyruvate, and then incubated at 37˚C for 10 weeks with weekly readings. Cultures were scored positive, negative, or contaminated. When contamination occurred, the original sample was reprocessed and reinoculated. Any suspected growth was tested for AFB using auramine O staining and fluorescence microscope as previously described [25]. All manipulations of samples including processing, inoculation, and DNA extraction were performed in the biosafety level three at National Reference Laboratory (NRL), Kigali, Rwanda.

## Molecular assays

**DNA extraction.** Lysate DNA was extracted from each AFB culture isolate as previously described [26]. Briefly, two loopful bacterial cells were suspended in 300 μl of distilled sterile water, then boiled at 95˚C for 25 min, quickly cooled, and stored at—20˚C until required. Genomic DNA was also extracted from inactivated grown cultures using a DNA extraction kit according to the manufacturer's instructions (Promega, USA).

**Conventional PCR.** Isolates that were confirmed AFB were screened for the presence of 16S rRNA sequence specific for the genus *Mycobacterium* and a sequence encoding the MPB 70 antigen which is specific for members of MTBC using specific primers (Mycgen–F and Mycgen-R, TB1-F and TB2-F, respectively) as previously described [27]. PCR assay based on genomic deletions differentiated members of MTBC (refer to as MTBC differential PCR assay) using primers targeting the regions of difference, RD1, RD4 and RD9 as previously described (Table 1) [28]. *Mycobacterium tuberculosis* 25177 was used as a reference. For all multiplex PCRs, the 15 μl PCR reaction mixture contained 1x MyTaq Red PCR Mix (Bioline, South Africa), 0.2 μM primers, and 2 μl of template DNA. The PCR cycling condition was as follows: initial denaturation at 94˚C for 5 min followed by 25 cycles of denaturation at 94˚C for 30 sec, annealing at 62˚C for 1 min, and extension at 72˚C for 1 min and a final extension step at 72˚C for 8 min. Primers amplified 1030bp for the genus, 372 bp for the MTBC, 108 bp and 268 bp for *M. bovis* as well as 108 bp and 235 bp for *M. tuberculosis*. PCR products were analysed by electrophoresis using a 2% agarose gel stained with red gel nucleic acid stain and fragments were visualized under UV light. The PCR experiments were performed at Rwanda Agriculture and Animal Resources Board, Department of Veterinary Services in the Virology and Molecular Biology sections.

**Table 1. Oligonucleotides used to identify *Mycobacterium* species isolated from slaughtered cattle in Rwanda.**

| PCRs | Primer name | Nucleotide sequence (5'-----------3') | Target | Gene or RD presence/ absence | Size (bp) | Tm (°C) | References |
|------|-------------|----------------------------------------|--------|------------------------------|-----------|---------|------------|
| **Multiplex 1** | Mycgen-F | AGA GTT TGA TCC TGG CTC AG | 16s rRNA | Gene is present in *M. bovis* and *M. tuberculosis* | 1030 | 62 | [27] |
| | Mycgen-R | TGC ACA CAG GCC ACA AGG GA | | | | | |
| | TB1-F | GAA CAA TCC GGA GTT GAC AA | MPB 70 | Gene is present in *M. bovis* and *M. tuberculosis* | 372 | | |
| | TB2-R | AGC ACG CTG TCA ATC ATG TA | | | | | |
| **Multiplex 2\*** | RD1-1 | AAGCGGTTGCCGCCGACCGACC | | RD1 present in *M. bovis* and *M. tuberculosis* | 146 | 62 | [29] |
| | RD1-2 | CTGGCTATATTCCTGGGCCCGG | | | | | |
| | RD1-3 | GAGGCGATCTGGCGGTTTGGGG | | | | | |
| | RD4-1 | ATG TGC GAG CTG AGC GAT G | Rv 1510 | RD4 absent in *M. bovis* but present in *M. tuberculosis* | 268 RD is absent | 62 | [28] |
| | RD4-2 | TGT ACT ATG CTG ACC CAT GCG | | | | | |
| | RD4-3 | AAA GGA GCA CCA TCG TCC AC | | | | | |
| | RD9-1 | CAA GTT GCC GTT TCG AGC C | Rv 2073 | RD9 absent in *M. bovis* but present *M. tuberculosis* | 108 RD is absent | 62 | [29] |
| | RD9-2 | CAA TGT TTG TTG CGC TGC | | | | | |
| | RD9-3 | GCT ACC CTC GAC CAA GTG TT | | | | | |

MPB 70 stands for protein from *M. bovis* with 0.70 mobility by native polyacrylamide gel electrophoresis at pH 9.4 gel but it is an antigen common to all MTBC, Rv refers to rough morphology and virulent MTBC strain and RD for regions of differences.

**GeneXpert/MTB/RIF assay.** MTBC isolates characterized by conventional PCR were also tested by GeneXpert/MTB/RIF molecular diagnostic assay following the manufacturer's instructions (Cepheid, Sunnyvale, USA). GeneXpert/MTB/RIF is a real-time PCR for the detection of MTBC and rifampin resistance. Briefly, 0.5 ml of the cell suspension was transferred into a conical-screwed tube and 1 ml of sample reagent was added. The mixture was vortexed for 10 sec and incubated for 15 min with vortexing for 10 sec after 8 min of incubation. The liquefied sample was then dispensed into the sample chamber of the cartridge containing five probes (A-E), integrated reagents tubes, a sample processing control, and a probe check control. Cartridges were then loaded into the GeneXpert Dx system version 4.8 (Cepheid, Sunnyvale, USA) and the amplification was run for two hours by activating the software installed in the computer. This assay was performed at NRL, Kigali, Rwanda.

## Data analysis

Data were recorded in the Microsoft Excel spreadsheet and descriptive and inferential statistics were performed using EpiInfo software version 7.2. The significance level of 95% and P-value less or equal to 5% were considered for all analyses. The prevalence of isolation of *Mycobacterium* and MTBC was tested for association with individual animal characteristics such as age, sex, breed, and sampled abattoir in the univariate logistic analysis using Chi-square or Fischer exact.

## Results

Out of the 300 sample pools collected from 300 cattle, 94.0% (282/300) were collected from the high throughput abattoirs while 6.0% (18/300) were from the low throughput abattoirs. Of the 300 sample pools, 95.3% (286, 95% CI: 92.3–97.4) were collected from female cattle while 4.7% (14, 95%CI: 2.6–7.7) were from male cattle. The majority of 90.3% (271/300, 95% CI: 86.4–93.4) were from adult (3 years and above) cattle while 9.7% (29/300, 95% CI: 6.6–13.6) were collected from young animals. Most sample pools, 67.9% (203/300, 95% CI: 62.3–73.2) were

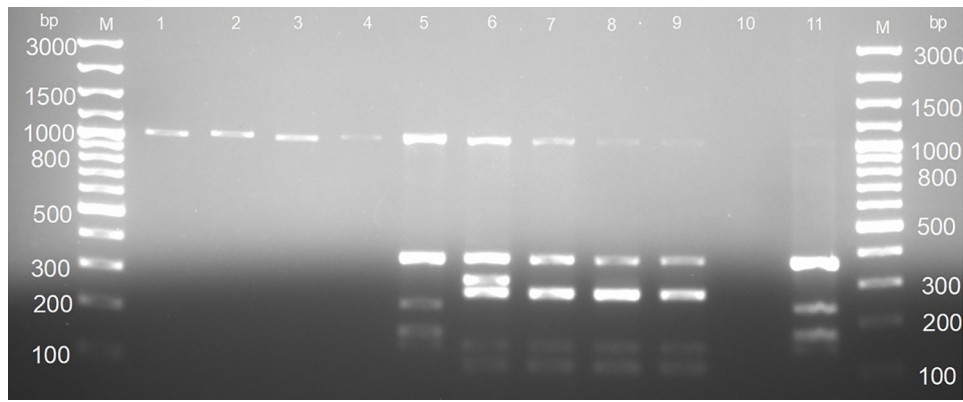

**Fig 2. Agarose gel electrophoresis of the *Mycobacterium tuberculosis* complex differential PCR assay.** Lane M: GeneRuler 100 bp (Invitrogen, Thermo Fischer Scientific, South Africa), lanes 1–4: Non-tuberculosis mycobacteria (NTM) which amplified 1030 bp; lane 5: *M. tuberculosis* which amplified 1030 bp, 372 bp, 235 bp, 172 bp, and 146 bp; lanes 6–9: *M. bovis* which amplified 1030 bp, 372 bp, 146 bp, and 108 bp; lane 10: sterile ultrapure water used as negative control; lane 11: *M. tuberculosis* reference strain 2517.

collected from crossbreeds, 25.8% (77/300, 95%CI: 20.9–31.1) were collected from local breed "Ankole", while 6.7% (20/300, 95%CI: 3.8–9.5) were collected from a pure breed "Friesian".

Of the 300 sample pools, six were collected with visible TB-like lesions including three sample pools with enlargement of all lymph nodes and one sample pool with granulomatous lesions in lungs. Other two sample pools had a mucoid pus in the popliteal lymph nodes. Of the 300 sample pools that were inoculated, 55.0% (165, 95% CI: 49.2–60.7) had bacterial growth of which 30.9% (51/165, 95% CI: 23.9–40.0) were AFB as indicated by the auramine fluorescence staining method. Of the 51 AFB, the PCR identified 80.4% (41/51, 95% CI: 69.5–91.3) as *Mycobacterium* spp. (amplification of 1030 bp fragment, Table 1). Of the 41 *Mycobacterium* spp., 87.8% (36/41, 95%CI: 77.8–97.8) were NTM, while the remaining 12.2% (5/41, 95% CI: 2.2–22.2) were MTBC (amplification of a 372 bp fragment, Table 1). MTBC differential PCR assay identified 80.0% (4/5, 95% CI: 44.9–100.0) as *M. bovis* (amplification of 108 bp, 146 bp, and 268 bp) and 20.0% (1/5, 95% CI: 0.0–55.1) as *M. tuberculosis* (amplification of 146 bp, 172 bp, 235 bp, 369 bp) (Fig 2). Overall, NTM were identified in 12.0% (36/300, 95%CI: 8.3–15.7) of the sampled cattle, MTBC were isolated in 1.7% (5/300) of the sampled cattle and among these, four were *M. bovis*, while one was *M. tuberculosis*. The GeneXpert MTBC/RIF assay confirmed MTBC isolates. Resistance to rifampicin (RR) was detected in one isolate, but such RR was not confirmed by GeneXpert testing of the original sample pool due to very low bacterial load.

Among the six sample pools that were collected with visible TB-like lesions, only one was confirmed as *M. bovis*, another one was NTM, while the remaining four were not identified as *Mycobacterium* spp.

*Mycobacterium* spp. isolates were found in 83.3% (5/6) of the abattoirs; 80.9% (38/47) in high throughput abattoirs and 75.0 (3/4) in low throughput abattoirs, although the difference was not significant (Table 2). All the MTBC isolates were found in the high throughput abattoirs. MTBC were isolated in cattle from Nyarugenge district (n = 1) of Kigali city, Karongi (n = 1), Nyabihu (n = 1), and Rubavu (n = 2) districts of the Western Province. All five MTBC isolates were identified from adult and crossbred cattle (Table 2).

## Discussion

Bovine tuberculosis (bTB) causes financial losses in livestock and remains a significant threat to public health worldwide [30,31]. This study aimed to determine the prevalence of bTB and

**Table 2. Mycobacterial culture results for 300 slaughtered cattle and PCR results of acid-fast bacilli isolates stratified by the abattoir, age, breed, and sex of slaughtered cattle in Rwanda.**

| Variables | Categories | Mycobacterial culture and AFB results | | PCR results of AFB isolates for detection of *Mycobacterium* spp. | | | | PCR results of AFB isolates for detection of *Mycobacterium tuberculosis* complex (MTBC) | | | |
|---|---|---|---|---|---|---|---|---|---|---|---|
| | | Growth % (n) | AFB positive | Positive % (n) | 95% CI | Chi $^2$ | *p*-value | Positive n (%) | 95% CI | Chi $^2$ | *p*-value |
| **Abattoirs** | High throughput | 53.9 (152/282) | 16.7 (47/282) | 80.9 (38/47) | 0.3–4.8 | 0.80 | 1.0 | 10.6 (5/47) | 0.0–16.3 | 0.00 | 1.00 |
| | Low throughput | 72.2 (13/18) | 22.2 (4/18) | 75.0 (3/4) | | | | 0.0 (0/4) | | | |
| | **Total** | **55.0 (165/300)** | **17.0 (51/300)** | **80.4 (41/51)** | | | | **9.8 (5/51)** | | | |
| **Provinces** | East | 55.7 (39/70) | 17.1 (12/70) | 83.3 (10/12) | - | 1.60 | 0.9 | 0.0 (0/12) | - | 10.20 | 0.04 |
| | West | 62.7 (44/70) | 25.7 (18/70) | 72.2 (13/18) | | | | 22.2 (4/18) | | | |
| | North | 56.0 (28/50) | 16.0 (8/50) | 87.5 (7/8) | | | | 0.0 (0/8) | | | |
| | South | 45.0 (36/80) | 13.8 (11/80) | 81.8 (9/11) | | | | 0.0 (0/11) | | | |
| | Kigali city | 60.0 (18/30) | 6.7 (2/30) | 100.0 (2/2) | | | | 50.0 (1/2) | | | |
| | **Total** | **55.0 (165/300)** | **17.0 (51/300)** | **80.4 (41/51)** | | | | **9.8 (5/51)** | | | |
| **Age** | Young | 65.5 (19/29) | 20.7 (6/29) | 50.0 (3/6) | 0.2–1.7 | 2.10 | 0.1 | 0.0 (0/6) | 0.0–9.3 | 0.02 | 1.00 |
| | Adult | 53.8 (146/271) | 16.6 (45/271) | 84.4 (38/45) | | | | 11.1 (5/45) | | | |
| | **Total** | **55.0 (165/300)** | **17.0 (51/300)** | **80.4 (41/51)** | | | | **9.8 (5/51)** | | | |
| **Sex** | Female | 55.6 (159/286) | 16.8 (48/286) | 79.2 (38/48) | - | 0.01 | 1.0 | 8.3 (4/48) | 0.4–74.8 | 0.20 | 0.30 |
| | Male | 42.9 (6/14) | 21.4 (3/14) | 100.0 (3/3) | | | | 33.3 (1/3) | | | |
| | **Total** | **55.0 (165/300)** | **17.0 (51/300)** | **80.4 (41/51)** | | | | **9.8 (5/51)** | | | |
| **Breeds** | Ankole | 54.5 (42/47) | 13.0 (10/77) | 80.0 (8/10) | - | 0.80 | 1.0 | 0.0 (0/10) | - | 1.90 | 0.70 |
| | Crossbred | 56.2 (114/208) | 18.7 (38/203) | 79.0 (30/38) | | | | 13.2 (5/38) | | | |
| | Friesians | 42.1 (8/19) | 15.0 (3/20) | 100.0 (3/3) | | | | 0.0 (0/3) | | | |
| | **Total** | **55.0 (165/300)** | **17.0 (51/300)** | **80.4 (41/51)** | | | | **9.8 (5/51)** | | | |

identify *Mycobacterium* species in slaughtered cattle using bacterial culture and PCR assay. Although the prevalence of bTB was low, the identification of *M. tuberculosis* with rifampicin resistance in cattle indicates the close contact between humans and cattle suggesting transmission of *M. tuberculosis* from humans to cattle. The identification of *M. bovis* in cattle suggests that farmworkers and abattoir workers can be diagnosed with zoonotic TB.

Considering the moderate sample size (n = 300) and the random selection of animals found without visible lesions (96.7%), the prevalence observed for MTBC in this study (1.7%) is higher than 0.5% reported at SABAN Nyabugogo abattoir, Rwanda [19] which was based on 36 samples from gross bTB-like lesions. However, considering that samples were collected at a single point in time and the small sample size (n = 10) per district, the prevalence in this study is likely to be lower than the true prevalence in slaughtered cattle in Rwanda. The prevalence (1.7%) obtained in this study is consistent with 2.1% obtained in Kenya using bovine TB-like lesions, AFB staining, bacterial culture, and PCR [32] but lower compared to 7.6% obtained in Uganda using bovine TB-like lesions, bacterial culture, AFB staining, and capillia TB-neo assay for detection of MPT 64 antigens of the MTBC [33]. The prevalence obtained in this study also falls in the range of bTB herd prevalence (0.2 to 13.2%) reported in cattle in Tanzania [34]. These findings confirm that bTB is present in cattle of the East African region and the disease remains a transboundary disease through animal trading and animal movement across porous borders [35].

This study demonstrated that bTB is most probably endemic in Rwandan cattle but at low prevalence since sample pools were collected from cattle slaughtered in the six abattoirs of which four high throughput abattoirs received cattle from several districts including those from more than 90 km. However, as mentioned in materials and methods section, only ten sample pools were collected from each of the 30 districts while MTBC isolates were detected in

animals from three districts. We, therefore, recommend a longitudinal study considering a larger sample size per district.

This study identified for the first time *M. bovis* and *M. tuberculosis* in slaughtered cattle in Rwanda. Similar studies in Africa isolated *M. bovis* and *M. tuberculosis* in slaughtered cattle in Kenya [32], in Cameroon [36]. However, the prevalence of *M. tuberculosis* in cattle is commonly below 1.0% [32,36,37], apart from some areas (27.0%) in Ethiopia where cattle owners had the habit of chewing tobacco into the mouth of their cattle [38]. The prevalence of *M. tuberculosis* is also high in areas with a high prevalence of TB in humans owning cattle [39]. For example, a study in the Eastern Cape province, South Africa, identified more *M. tuberculosis* (41.8%, 157/376) than *M. bovis* (1.3%, 5/376) from slaughtered cattle [40]. *Mycobacterium tuberculosis* (n = 1) that was isolated in this study was resistant to Rifampicin which is a cornerstone antibiotic of the first-line regimen [41]. Resistance to rifampicin (RR) is considered multidrug resistance tuberculosis (MDR-TB) [42] and has been reported in Rwandans with TB [43] suggesting that RR *M. tuberculosis* was of human origin. The transmission of RR *M. tuberculosis* from humans to cattle was not surprising since 92.0% of Rwandans owning cattle are small dairy farmers practicing a zero-grazing system [44] which may promote close animal-human contact with the risk of cross-infection from humans to cattle and vice-versa [45]. Cattle can be considered sentinels for *M. tuberculosis* in settings where TB is not effectively controlled in humans. It is therefore also an alert for improved TB control in humans in rural settings.

Despite the little attention given to the zoonotic TB caused by *M. bovis* [46], several studies have isolated *M. bovis* in extrapulmonary lymph nodes of humans in neighbouring Uganda [47], and Tanzania [48]. It is hence essential to raise awareness among veterinary and human health professionals about the zooanthroponotic and anthropozoonotic transmission of TB in Rwanda. Further studies on the identification of *M. bovis* in occupational groups are worth investigating in Rwanda to provide epidemiological data that are indispensable for the eradication of tuberculosis by 2035 [49].

The prevalence of NTM (12.0%) in this study is consistent with 8.4% obtained in Uganda [33] but higher than 3.9% in Tanzania [13]. This study considered the presence of NTM in the environment, hence, tissues were aseptically (changing gloves and sterilization of the knife into hot water) collected, stored, and processed, thus, it can be assumed that the identified NTM were recovered from the tissues of animals, but it does not prove any pathological effect, it merely demonstrates colonization. Since these NTM were not speciated, it would be important to determine their potential significance for the health of the cattle. NTM have been isolated in cattle and sometimes cause localized lymphadenitis, skin infections, TB-like pulmonary infections, and systemic diseases in immunodeficient cattle [50]. The presence of NTM in cattle may interfere with immune-diagnostic methods such as comparative tuberculin test and may negatively impact vaccination [51].

In this study, among the six sample pools with tuberculosis-like lesions, only one popliteal lymph node was associated with NTM species, and one lung was associated with *M. bovis*. However, a retrospective study reported the prevalence of 11.8% based on TB-like lesions recorded during routine meat inspection from 2006 to 2010 at SABAN Nyabugogo abattoir, Rwanda [20]. TB-like lesions might therefore be a poor reflection of bTB in the absence of a confirmatory laboratory test. TB-like lesions from routine meat inspection should, therefore, be confirmed by laboratory tests to obtain accurate results essential for surveillance of bTB, but also improve the knowledge of inspectors.

*Mycobacterium* spp. isolates were more frequently isolated in adult than in young cattle and all MTBC were isolated from adult cattle consistent with a study in Ethiopia [52]. The isolation of *M. bovis* depends a lot on the dose and frequency of exposure, therefore, higher infection

rates in adult cattle result from a cumulative risk of infection. In other words, the older an animal the more opportunities it had to contract *M. bovis*. Furthermore, the literature states that young cattle are less susceptible to mycobacteria owing to the high concentration of T cells in the blood circulation and T cells play a role in the immunity against mycobacteria [53].

## Conclusions

This study demonstrated that bTB is present in Rwanda at low prevalence. The present study reports for the first time MTBC in cattle in Rwanda and the presence of RR *M. tuberculosis* indicating possible cross-infection between humans and cattle. There is therefore a need for raising awareness among veterinary and human health professionals about the zooanthroponotic transmission and cross infection of TB in Rwanda. Further studies on the identification of *M. bovis* in humans are worth investigating to provide epidemiological data that are indispensable for the eradication of tuberculosis by 2035, a global movement led by the World Health Organization.

## Supporting information

**S1 Data. Raw data.** Excel file.
(XLSX)

## Acknowledgments

The authors would like to acknowledge the National Reference Laboratory (NRL) and the University of Rwanda for the facilitation of the study. We thank also NRL staff in the tuberculosis section, veterinary inspectors of the sampled abattoirs, abattoir managers, and abattoirs workers for their good cooperation.

## Author Contributions

**Conceptualization:** Jean Bosco Ntivuguruzwa, Anita L. Michel, Henriette van Heerden.

**Data curation:** Jean Bosco Ntivuguruzwa.

**Formal analysis:** Jean Bosco Ntivuguruzwa.

**Funding acquisition:** Henriette van Heerden.

**Investigation:** Jean Bosco Ntivuguruzwa.

**Methodology:** Jean Bosco Ntivuguruzwa, Anita L. Michel, Henriette van Heerden.

**Project administration:** Henriette van Heerden.

**Resources:** Ivan Emil Mwikarago, Jean Claude Semuto Ngabonziza, Henriette van Heerden.

**Software:** Jean Bosco Ntivuguruzwa.

**Supervision:** Anita L. Michel, Francis Babaman Kolo, Henriette van Heerden.

**Validation:** Anita L. Michel.

**Writing – original draft:** Jean Bosco Ntivuguruzwa.

**Writing – review & editing:** Jean Bosco Ntivuguruzwa, Anita L. Michel, Francis Babaman Kolo, Ivan Emil Mwikarago, Jean Claude Semuto Ngabonziza, Henriette van Heerden.

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
