## [Decision Letter · Decision Letter 0]

14 Mar 2022

Dear Dr. Ntivuguruzwa,

Thank you very much for submitting your manuscript "Prevalence of bovine tuberculosis and characterization of the members of the Mycobacterium tuberculosis complex from slaughtered cattle in Rwanda" for consideration at PLOS Neglected Tropical Diseases. As with all papers reviewed by the journal, your manuscript was reviewed by members of the editorial board and by several independent reviewers. The reviewers appreciated the attention to an important topic. Based on the reviews, we are likely to accept this manuscript for publication, providing that you modify the manuscript according to the review recommendations. 

Sincerely,

Subash Babu

Associate Editor

Ana LTO Nascimento

Deputy Editor

Reviewer's Responses to Questions

**Key Review Criteria Required for Acceptance?**

**Methods**

-Are the objectives of the study clearly articulated with a clear testable hypothesis stated?

-Is the study design appropriate to address the stated objectives?

-Is the population clearly described and appropriate for the hypothesis being tested?

-Is the sample size sufficient to ensure adequate power to address the hypothesis being tested?

-Were correct statistical analysis used to support conclusions?

-Are there concerns about ethical or regulatory requirements being met?

Reviewer #1: The research study was well introduced and sufficient. The aim and objectives were also well presented and to the point. The authors demonstrated knowledge and understanding of the nature and purpose of the subject matter. Sample size and analysis of the analysis for the purpose of the study were determined statistically. Necessary ethical clearance was obtained before the study was commenced.

Reviewer #2: It will be essential to have an additional confirmation test for the diagnosed NTMs. (any other existing biochemical or microbiological confirmation, other than culture and AFB staining, can be done to substantiate their findings, other than this PCR method.

This is essential for the validity for their results.

**Results**

-Does the analysis presented match the analysis plan?

-Are the results clearly and completely presented?

-Are the figures (Tables, Images) of sufficient quality for clarity?

Reviewer #1: Results obtained were statistically analyzed and have been presented in ways that are clear, logical and appropriate for the purpose. However, special attention needs to be given to the gel electrophoresis picture (Figure 1). Some lanes are missing (labelling) hence affecting the correctness of the information provided in the figure legend.

Reviewer #2: The gel picture in the results section does not have the lane 4 mentioned, as NTM and hence request them to provide the corrected gel picture, with proper labeling. many mistakes in the labeling and legends needs to corrected. Negative control lane has bands, hence negative for what?

Why not the WGS approach for the isolated strains carried out. 

additional information with reference to the lineages can be added for more authenticity of the data.

**Conclusions**

-Are the conclusions supported by the data presented?

-Are the limitations of analysis clearly described?

-Do the authors discuss how these data can be helpful to advance our understanding of the topic under study?

-Is public health relevance addressed?

Reviewer #1: The study is first to confirm Mycobacterium bovis, Mycobacterium tuberculosis and non-tuberculous mycobacteria in Rwanda. Additional interesting finding from the study was that not all TB-like lesions observed resulted in the isolation of Mycobacterium species, highlighting the importance of laboratory confirmation of the causative agent to obtain accurate diagnosis. Considering the moderate sample size analyzed, authors’ recommended future longitudinal studies which will consider a good sample size per district in that country. It will also be worthwhile to genotype MTBC identified to determine the genetic diversity of the strains circulating in Rwanda for epidemiological studies. The study found presence of Rifampicin resistant M. tuberculosis indicating possible cross-infection between humans and cattle. Authors highlighted a need for raising awareness among veterinary and human health professionals about the zooanthroponotic transmission and cross infection of TB in Rwanda.

Reviewer #2: This study will become valid with their additional experiments to prove the NTMs and M.bovis. It will also good for the authors to mention about the implications of the study carried out in the discussion, emphasizing how this will be helpful for the animal husbandry and zoonosis transmission besides prevalence.

**Editorial and Data Presentation Modifications?**

Reviewer #1: Additional comments are highlighted in the attachment.

Reviewer #2: Gel picture needs extensive revision, may be relabeling or a new gel picture required .

**Summary and General Comments**

Reviewer #1: This work aimed to determine the prevalence of bovine tuberculosis and distinguish amongst members of the Mycobacterium tuberculosis complex (MTBC) species. The study is first to confirm Mycobacterium bovis, Mycobacterium tuberculosis and non-tuberculous mycobacteria in Rwanda. Additional interesting finding from the study was that not all TB-like lesions observed resulted in the isolation of Mycobacterium species, highlighting the importance of laboratory confirmation of the causative agent to obtain accurate diagnosis. Considering the moderate sample size analyzed, authors’ recommended future longitudinal studies which will consider a good sample size per district in that country. It will also be worthwhile to genotype MTBC identified to determine the genetic diversity of the strains circulating in Rwanda for epidemiological studies.

Reviewer #2: This study involves identification of MTBC members to understand transmission dynamics among animal-human interface and to design adequate control programs to prevent the same.

PLOS authors have the option to publish the peer review history of their article (what does this mean?). If published, this will include your full peer review and any attached files.

Reviewer #1: No

Reviewer #2: No

Figure Files:

Data Requirements:

Reproducibility:

References

---

## [Decision Letter · Decision Letter 1]

22 Jun 2022

Dear Dr. Ntivuguruzwa,

Thank you very much for submitting your manuscript "Prevalence of bovine tuberculosis and characterization of the members of the Mycobacterium tuberculosis complex from slaughtered cattle in Rwanda" for consideration at PLOS Neglected Tropical Diseases. As with all papers reviewed by the journal, your manuscript was reviewed by members of the editorial board and by several independent reviewers. The reviewers appreciated the attention to an important topic. Based on the reviews, we are likely to accept this manuscript for publication, providing that you modify the manuscript according to the review recommendations. 

Sincerely,

Subash Babu

Associate Editor

Ana LTO Nascimento

Deputy Editor

Reviewer's Responses to Questions

**Key Review Criteria Required for Acceptance?**

**Methods**

-Are the objectives of the study clearly articulated with a clear testable hypothesis stated?

-Is the study design appropriate to address the stated objectives?

-Is the population clearly described and appropriate for the hypothesis being tested?

-Is the sample size sufficient to ensure adequate power to address the hypothesis being tested?

-Were correct statistical analysis used to support conclusions?

-Are there concerns about ethical or regulatory requirements being met?

Reviewer #1: Overall, the manuscript is well written and scientifically sound. The aim and objectives are clearly articulated and to the point. Well known statistical methods were used to determine the sample size and results analysis. The study was approved by relevant ethics committees.

Reviewer #2: (No Response)

**Results**

-Does the analysis presented match the analysis plan?

-Are the results clearly and completely presented?

-Are the figures (Tables, Images) of sufficient quality for clarity?

Reviewer #1: Overall, results were presented well. However, minor additions/clarity is needed in table 1 and figure 1.

Reviewer #2: The essential changes in the gel picture have been carried out and the response to the queries has been carried out satisfactorily.

**Conclusions**

-Are the conclusions supported by the data presented?

-Are the limitations of analysis clearly described?

-Do the authors discuss how these data can be helpful to advance our understanding of the topic under study?

-Is public health relevance addressed?

Reviewer #1: (No Response)

Reviewer #2: (No Response)

**Editorial and Data Presentation Modifications?**

Reviewer #1: This is a cross sectional study that utilized a systematic sampling of cattle as well as mycobacterial species culture and molecular based techniques to determine the prevalence of bovine tuberculosis and characterize members of the isolated Mycobacterium tuberculosis complex (MTBC). The manuscript was well written and it is scientifically sound.

Comments:-

Introduction

Line 65…. It will be worthwhile to mention other members of the MTBC that cause bovine tuberculosis besides M. bovis and M. caprae before mentioning that M. tuberculosis infection in cattle has been identified more recently in African countries….

Conventional PCR

Line 179-180…Authors should also mention the fragment sizes expected for the identification of M. tuberculosis….

GeneXpert MTB/RIF assay

Line 192-193… Cartridges were ‘installed’ into the Gene Xpert DX system… Please replace ‘installed’ with ‘loaded’….

Table 1

Authors should make is clear which target gene sequence is present/absent in which mycobacterium species (i.e. RD4 is absent in M. bovis but present in M. tuberculosis etc…)

Figure 2

Line 250… Authors should please indicate what was used as a negative control for quality control purposes.

Discussion

Line 162.. Remove ‘Therefore’ at the beginning of the sentence as this point does not relate to the previous one.

Reviewer #2: Data has been represented in a better manner

**Summary and General Comments**

Reviewer #1: Indicated in the section above.

Reviewer #2: Can be considered for publication

PLOS authors have the option to publish the peer review history of their article (what does this mean?). If published, this will include your full peer review and any attached files.

Reviewer #1: No

Reviewer #2: Yes: Ramalingam Bethunaickan

Figure Files:

Data Requirements:

Reproducibility:

References

---

## [Decision Letter · Decision Letter 2]

8 Jul 2022

Dear Dr. Ntivuguruzwa,

We are pleased to inform you that your manuscript 'Prevalence of bovine tuberculosis and characterization of the members of the Mycobacterium tuberculosis complex from slaughtered cattle in Rwanda' has been provisionally accepted for publication in PLOS Neglected Tropical Diseases.

Best regards,

Subash Babu

Associate Editor

Ana LTO Nascimento

Deputy Editor

Reviewer's Responses to Questions

**Key Review Criteria Required for Acceptance?**

**Methods**

-Are the objectives of the study clearly articulated with a clear testable hypothesis stated?

-Is the study design appropriate to address the stated objectives?

-Is the population clearly described and appropriate for the hypothesis being tested?

-Is the sample size sufficient to ensure adequate power to address the hypothesis being tested?

-Were correct statistical analysis used to support conclusions?

-Are there concerns about ethical or regulatory requirements being met?

Reviewer #1: Yes.

**Results**

-Does the analysis presented match the analysis plan?

-Are the results clearly and completely presented?

-Are the figures (Tables, Images) of sufficient quality for clarity?

Reviewer #1: Yes.

**Conclusions**

-Are the conclusions supported by the data presented?

-Are the limitations of analysis clearly described?

-Do the authors discuss how these data can be helpful to advance our understanding of the topic under study?

-Is public health relevance addressed?

Reviewer #1: Yes.

**Editorial and Data Presentation Modifications?**

Reviewer #1: Accept.

**Summary and General Comments**

Reviewer #1: Authors satisfactorily responded to all comments made.

PLOS authors have the option to publish the peer review history of their article (what does this mean?). If published, this will include your full peer review and any attached files.

Reviewer #1: No

---

## [Editor Report · Acceptance letter]

30 Jul 2022

Dear Dr Ntivuguruzwa,

We are delighted to inform you that your manuscript, " Prevalence of bovine tuberculosis and characterization of the members of the Mycobacterium tuberculosis complex from slaughtered cattle in Rwanda ," has been formally accepted for publication in PLOS Neglected Tropical Diseases.

Best regards,

Shaden Kamhawi

co-Editor-in-Chief

Paul Brindley

co-Editor-in-Chief
